# Direct observation of trap-assisted recombination in organic photovoltaic devices

Stefan Zeiske[1], Oskar J. Sandberg [1✉], Nasim Zarrabi[1], Wei Li[1], Paul Meredith [1] & Ardalan Armin [1✉]

Trap-assisted recombination caused by localised sub-gap states is one of the most important first-order loss mechanism limiting the power-conversion efficiency of all solar cells. The presence and relevance of trap-assisted recombination in organic photovoltaic devices is still a matter of some considerable ambiguity and debate, hindering the field as it seeks to deliver ever higher efficiencies and ultimately a viable new solar photovoltaic technology. In this work, we show that trap-assisted recombination loss of photocurrent is universally present under operational conditions in a wide variety of organic solar cell materials including the new non-fullerene electron acceptor systems currently breaking all efficiency records. The trap-assisted recombination is found to be induced by states lying 0.35-0.6 eV below the transport edge, acting as deep trap states at light intensities equivalent to 1 sun. Apart from limiting the photocurrent, we show that the associated trap-assisted recombination via these comparatively deep traps is also responsible for ideality factors between 1 and 2, shedding further light on another open and important question as to the fundamental working principles of organic solar cells. Our results also provide insights for avoiding trap-induced losses in related indoor photovoltaic and photodetector applications.

[1] Sustainable Advanced Materials (Sêr-SAM), Department of Physics, Swansea University, Singleton Park, Swansea SA2 8PP Wales, UK. ✉email: o.j.sandberg@swansea.ac.uk; ardalan.armin@swansea.ac.uk

Recently, organic solar cells have surpassed 17%[1,2] power conversion efficiency (PCE) in single-absorber layer bulk heterojunction (BHJ) devices based upon non-fullerene electron acceptor systems. This represents a major advance in the field, and indeed may be the catalyst to move organic solar cells to 'viable technology' status. BHJs have also been used for indoor applications[3] as well as in state-of-the-art photodetectors, such as photodiodes with wide dynamic range[4], high specific detectivities[5,6], and color selectivity[7]. In order to further optimize the performance of organic photovoltaic devices, including organic solar cells, indoor cells, and photodetectors, a better understanding of fundamental processes limiting the photocurrent and thereby the PCE, is a pre-requisite. In state-of-the-art devices, the short-circuit current is limited by first-order losses, including those due to non-optimal absorption and geminate recombination of excitons and charge transfer (CT) states. Another important first-order loss mechanism is recombination via trap states – that is, through available states for charge carriers within the energy gap. In general, traps and the associated trap-assisted recombination also give rise to increased non-radiative photovoltage losses and reduced fill factors. While losses caused by non-optimal absorption and geminate recombination have been studied in detail and can be directly linked to material properties, the presence and role of traps remain controversial topics in organic solar cells and organic optoelectronics more broadly.

In neat homojunction organic devices, universal traps at 3.6 and 6.0 eV (below the vacuum level) limiting charge carrier transport have been observed, attributed to water-oxygen complexes and water clusters, respectively[8,9]. These traps define an energetic window for bipolar trap-free charge transport in organic semiconductors: if the lowest unoccupied molecular orbital (LUMO) [or highest occupied molecular orbital (HOMO)] lies above [below] the electron [hole] trap level, the electron [hole] transport is inevitably trap-limited. This explains the unipolarity of long-range charge transport in most organic semiconductors at low carrier densities[8,10]. However, we note that a different conclusion was reached in a recent study by Zuo et al. who suggested that the traps are caused by water-filled nanovoids, inducing trap levels that are always ~0.3–0.4 eV above (below) the HOMO (LUMO)[11].

In BHJs, the LUMO of the electron-transporting acceptor and the HOMO of the hole-transporting donor in the active layer typically lie within the 3.6–6.0 eV window of trap-free electron and hole transport in their respective domains. Despite this, the presence of traps (and trap-assisted recombination) has been frequently reported for organic BHJ solar cells[12–20]. While the debate has been heavily centered around whether these traps are mid-gap states or shallow tail states below the transport level, surprisingly little is known about their energetics. Furthermore, the actual trap-induced losses under realistic operating conditions (i.e., intensities of ~1 sun and at maximum power point) have in many cases remained matters of conjecture—the nature and presence of trap-assisted recombination are frequently deduced from ideality factor measurements or electrical transient methods, such as transient photovoltage, performed under open-circuit conditions. However, these methods have been shown to be susceptible to both electrode-induced and resistive/transport-related limitations[21–25]. To date, reliable methods that unambiguously quantify photocurrent losses caused by trap-assisted recombination in organic solar cells under relevant operating conditions are lacking.

Herein, we utilize wide dynamic range light intensity-dependent measurements of the external quantum efficiency (EQE) at sub-Hz modulation frequencies to characterize trap-assisted recombination in fully operational organic photovoltaic devices. We show that in the presence of traps, an anomalous and hitherto unreported two-step EQE behavior can be observed (as a function of incident light intensity). The two-step behavior is a result of trap-induced first-order recombination in the bulk (which is absent at low intensities) being switched on at moderate intensities due to trap filling. This effect allows for the first-order trap-assisted recombination losses under 1 sun intensity to be quantified. Combined intensity-dependent photocurrent (IPC) and open-circuit voltage ($V_{oc}$) measurements further allow estimates of the energetic depth of trap states within the active layer. Using this method, we find that trap-induced first-order photocurrent losses are present in a large variety of fullerene and non-fullerene-based organic BHJ solar cells—a sufficient enough set of material systems to consider the observation universal. This first-order recombination loss is caused by traps states in the gap, lying in all cases 0.35–0.6 eV below the transport levels. Finally, our findings are also relevant for indoor light harvesting and photodetector applications of organic BHJs operating at low intensities.

## Results

In Fig. 1 the basic principle behind our analysis method is demonstrated. Figure 1a shows a schematic energy level diagram for a BHJ with electron trap states of density $N_t$ situated at an energy $E_t$ in the gap. The trap depth is then defined as $\Delta_t = E_{L,A} - E_t$, where $E_{L,A}$ is the LUMO level energy of the acceptor. In accordance with the Shockley–Read–Hall (SRH) formalism, the associated trap-assisted recombination takes place between trapped electrons (captured from the LUMO) and free holes. Figures 1b, c show the simulated normalized EQE and $V_{oc}$ vs. light intensity ($I_L$) for a generic organic solar cell with and without traps. A numerical drift-diffusion model was used for the simulations;[26] the details of which are provided in the Supplementary Information. For the device with traps, the simulations reveal a constant EQE at low intensities and a second semi-plateau at higher intensities (albeit with lower magnitude) caused by the first-order trap-assisted recombination in the bulk. For the trap-free device, the second plateau is notably absent.

The SRH rate is limited by the number of trapped electrons, which depends on the position of the quasi-Fermi level $E_{F,n}$ for free electrons relative to the trap energy $E_t$. The carrier density, and thus the quasi-Fermi level, in turn, depends on the incident light intensity. Under conditions when $E_{F,n} < E_t$ (low $I_L$, see inset in Fig. 1b), most of the traps are unoccupied (shallow trap mode), and the trap-assisted recombination is negligible (compared to charge extraction). In contrast, when $E_{F,n} > E_t$ (high $I_L$, see inset in Fig. 1b), a significant fraction of traps will be occupied by electrons (deep trap mode), and first-order trap-assisted recombination is switched on, resulting in the second EQE plateau. The critical electron density at which $E_{F,n} = E_t$ is given by

$$n_1 = N_{L,A} \exp\left(-\frac{\Delta_t}{kT}\right), \tag{1}$$

where $N_{L,A}$ is the density of transport states in the LUMO of the acceptor, $k$ is the Boltzmann constant and $T$ is the absolute temperature. Hence, the trap depth critically defines the onset intensity for first-order SRH recombination in the bulk—below this onset, the traps act as shallow traps and the trap-induced first-order recombination losses in the bulk are small. Finally, at even higher intensities second-order bimolecular recombination eventually starts to play a role, manifested as an additional light intensity-dependent quantum efficiency (QE) loss.

We note that the competition between trap-assisted recombination and bimolecular recombination is more pronounced under open-circuit conditions. The corresponding ideality factor $n_{id}$ is related to the open-circuit voltage via $V_{oc} = (n_{id}kT/q) \times \ln(I_L) + \text{constant}$, where $q$ is the elementary charge. As evident

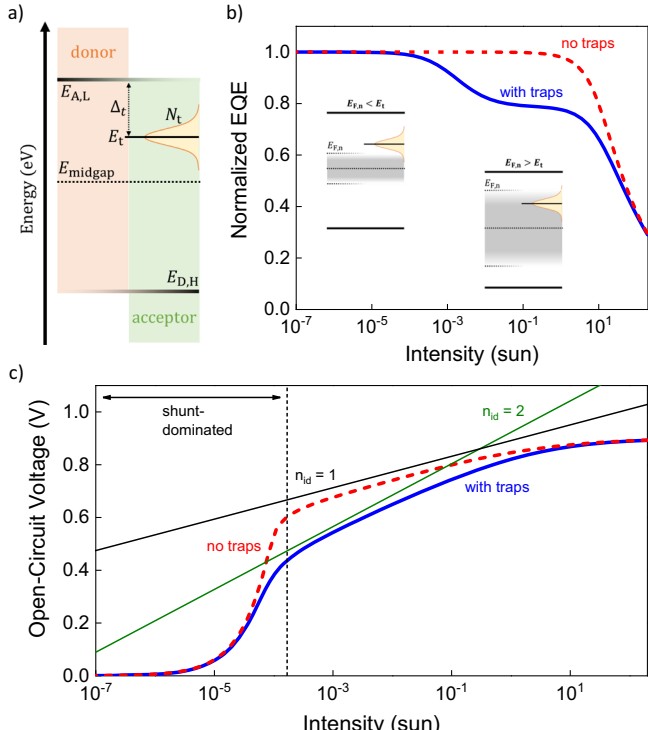

**Fig. 1 Schematic energy level diagram and drift-diffusion simulations of intensity-dependent EQE and $V_{oc}$ in the presence of trap states. a** Schematic energy level diagram of donor: acceptor BHJ solar cell with acceptor LUMO level ($E_{A,L}$) and donor HOMO level ($E_{D,H}$). The electron trap is located at an energy $E_t$ having a trap depth of $\Delta_t = |E_{A,L} - E_t|$ and trap density $N_t$. **b** Comparison of simulated external quantum efficiency (EQE) vs. intensity between the case with (blue, solid line) and without (red, dashed line) trap states in the donor: acceptor blend bulk. In the presence of comparably deep trap states a unique two-step EQE is seen at low and moderate intensities. The insets depict the schematic energy level diagrams for the corresponding intensity regime, where $E_{F,n} < E_t$ at low intensity and $E_{F,n} > E_t$ at moderate intensity. **c** The open-circuit voltage simulated as a function of light intensity, and compared for the case with (blue, solid line) and without (red, dashed line) trap states. The black and green solid lines are guides to the eye with a slope corresponding to $n_{id} = 1$ (i.e., without traps) and $n_{id} = 2$ (i.e., with traps). The vertical, dotted line marks the intensity below which the open-circuit voltage is shunt-dominated.

from Fig. 1c, while the $V_{oc}$ at low and very high intensities is limited by shunt effects ($n_{id} > 2$) and the contacts ($n_{id} < 1$)[27], respectively, the $V_{oc}$ at moderate intensities is dominated by bulk recombination. Depending on the relative balance between the first-order trap-assisted recombination ($n_{id} = 2$) and bimolecular recombination ($n_{id} = 1$), the ideality factor transitions from 2 to 1 resulting in arbitrary values between 1 and 2 when varying the photovoltage (through control of the incident light intensity).

In the following, we use intensity-dependent EQE measurements to experimentally quantify the QE loss caused by trap-assisted recombination in the bulk. For this purpose, we conduct IPC measurements under short-circuit conditions at an excitation wavelength of 520 nm. The associated two-step EQE behavior is distinctly different to that observed for other first-order loss mechanisms (e.g., of CT states and excitons) which are expected to be independent of light intensity. Thus, the presence of such a two-step EQE can be used to unambiguously identify trap-assisted recombination in organic photovoltaics devices. We note that to experimentally detect first-order recombination via trap

states, the photocurrent needs to be recorded over a broad intensity range and at extremely low modulation frequencies ($f$) so that $1/f$ is longer than the trap release time. Due to the dominant impact of flicker noise at low frequencies, such measurements typically require long integration times (here more than 30 s for each intensity point at low intensities).

Figures 2a, b show the light intensity-dependent normalized EQE and $V_{oc}$, respectively, of three different organic solar cells: PCDTBT:PC$_{70}$BM, PTB7-Th:PC$_{70}$BM and PM6:BTP-eC9. The chemical definitions are provided in the Supplementary Information. All devices exhibit a constant EQE plateau at low intensities reaching a second semi-plateau at moderate intensity with lower magnitude, consistent with the presence of trap-assisted recombination in the bulk (Fig. 1). Corresponding first-order, trap-induced, relative QE losses of 5%, 4%, and 3% were obtained from the IPC measurements for the three systems, respectively. However, this loss channel is deactivated below intensities around $10^{-3}$–$10^{-4}$ suns for PCDTBT:PC$_{70}$BM and PTB7-Th:PC$_{70}$BM, and $10^{-2}$ suns for PM6:BTP-eC9. The presence of trap-assisted recombination is further corroborated by the $V_{oc}$ data with ideality factors varying between 1 and 2 at moderate intensities. At intensities above 1 sun equivalent, higher-order processes (such as bimolecular recombination and series resistance limitations) eventually lead to a rapid decrease of the EQE.

This two-step EQE behavior was detected for a large variety of fullerene and non-fullerene acceptor-based organic solar cells (see Supplementary Fig. 4). Details of the device fabrication and current density-voltage performances measurements under artificial AM 1.5G conditions are provided in the Supplementary Information (see Supplementary Fig. 3 and 10 as well as Supplementary Table 3 and 4). The associated relative QE losses for the studied systems are summarized in Fig. 2c. We note that the relative QE losses are not only an indication of how much (relative) photocurrent is lost due to the first-order, trap-assisted recombination channels, but also how much (relative) more photocurrent could be gained in the absence of SRH recombination pathways. The corresponding slope parameter $\alpha$ of the short-circuit density, $J_{sc} \propto I_L^\alpha$, in the intensity regime of the second EQE semi-plateau are shown in Fig. 2d. For all systems, an $\alpha$ close to unity was obtained, indicating that the photocurrent loss associated with the second plateau is indeed first-order. Although we cannot realistically study every possible combination of acceptor and donor organic semiconductors, the consistency of these findings points towards the universal presence of first-order recombination via deep traps in the bulk of organic solar cells.

To estimate the associated trap depth, we make use of Eq. 1. Accordingly, the trap depth can be obtained from the average free carrier density ($n = n_1$) at the point-of-transition (POT) intensity right in-between the two EQE plateaus. The photogeneration current at POT is given by $J_{G,POT} = 2qnd/t_{tr}$, where $d$ is the active layer thickness and $t_{tr}$ is the carrier transit time. Hence, the trap depth can be estimated via $\Delta_t = kT \ln(2qN_{L,A}d/[J_{G,POT}t_{tr}])$ (Supplementary Note 1). Figure 2e shows the corresponding trap depths for the different systems. The transit times were obtained from resistance-dependent photovoltage (RPV) measurements (see Supplementary Fig. 7)[28,29]. The trap depths are all between 0.35 and 0.6 eV for both fullerene and non-fullerene systems. We note that the extracted trap depths are consistent with those experimentally expected from the measured open-circuit voltages (see Supplementary Note 2).

The experimental EQE and $V_{oc}$ (and corresponding J-V curves under 1 sun) can be reproduced qualitatively by drift-diffusion simulations, assuming a device with trap states lying 0.4–0.5 eV below the transport levels and a finite shunt resistance, as indicated by the solid lines in Fig. 2a, b (and Supplementary Fig. 9).

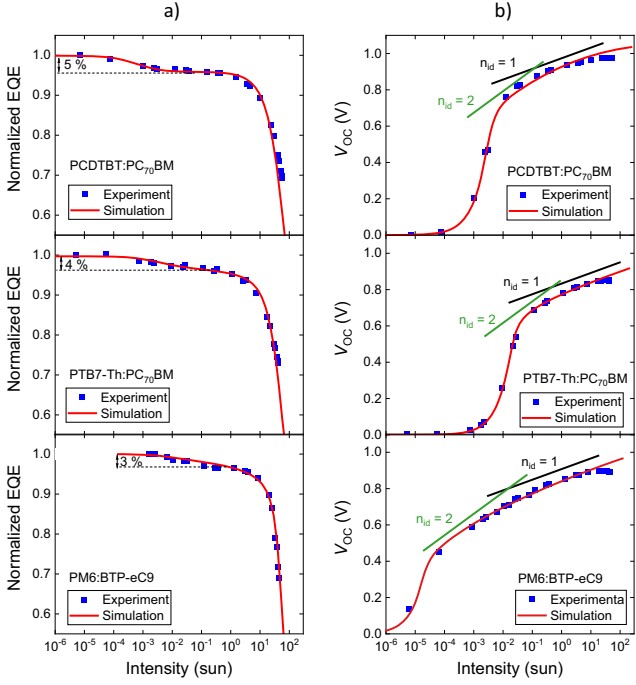

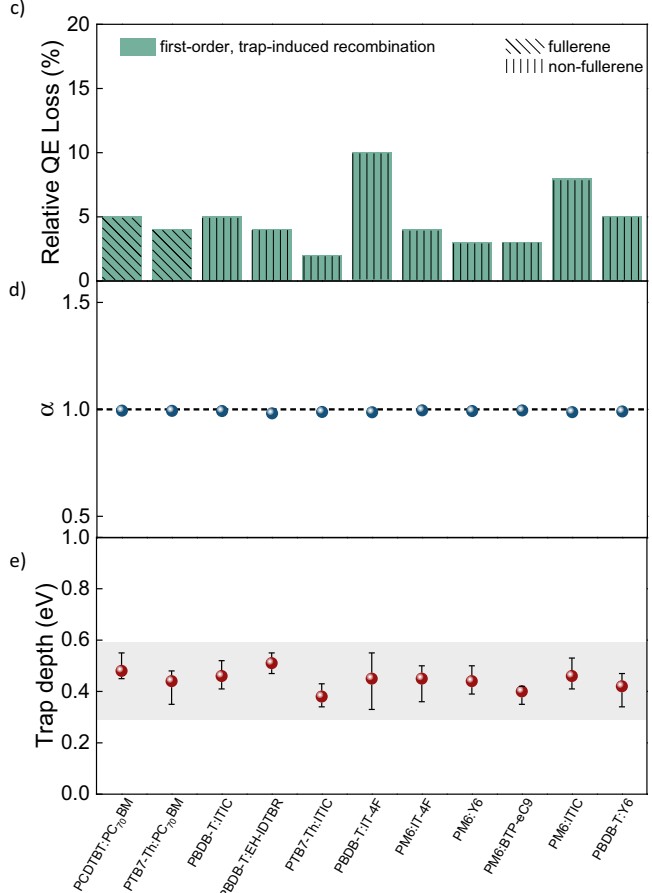

**Fig. 2 First-order, trap-induced recombination in fullerene and non-fullerene acceptor-based organic solar cells. a** Experimental (blue symbols) and simulated (red lines) normalized external quantum efficiency (EQE) plotted as a function of intensity for three different organic solar cells together with the estimated relative QE losses induced by first-order, trap-assisted recombination: PCDTBT:PC$_{70}$BM (5%), PTB7-Th:PC$_{70}$BM (4%) and PM6:BTP-eC9 (3%). **b** Corresponding open-circuit voltage ($V_{oc}$) plotted as a function of intensity for the three solar cells. The black and green lines are guides to the eye with a slope corresponding to an ideality factor of $n_{id} = 2$ (trap-assisted recombination is dominant) and $n_{id} = 1$ (bimolecular recombination dominates). **c** Relative QE loss induced by first-order, trap-assisted recombination for different fullerene and non-fullerene acceptor-based organic solar cells. The QE losses were determined from IPC measurements. **d** Slope parameter $\alpha$ extracted from within the second EQE plateau for a large variety of donor: acceptor BHJ systems. The horizontal, dotted line marks the slope $\alpha = 1$ corresponding to a first-order process, consistent with deep gap states at these intensities. **e** The corresponding estimated trap depths for the different organic solar cell systems from sensitive IPC measurements performed over a broad range of intensities. Trap depths scatter between 0.35 and 0.6 eV. The error bars correspond to maximum/minimum calculated trap depth from lower/higher end of transition regime between the two EQE plateaus.

factors in organic solar cells. An estimate of the active trap density $N_t$ can be obtained noting that occupied traps, similar to the case of doping or imbalanced mobilities[30,31], will induce a space-charge region inside the active layer limiting the photo-current (see Supplementary Note 1). Based on this estimate, the trap density for the studied BHJ systems is found to be on the order of $10^{16}$–$10^{17}$ cm$^{-3}$.

Our results are consistent with the presence of trap-assisted recombination that is activated at moderate intensities by trap filling of states, lying 0.35–0.6 eV below the transport levels, acting as deep trap states under 1 sun illumination. The onset intensity, at which the first-order trap-assisted recombination in the bulk is activated, is determined by the trap depth. Con-comitantly, the critical trap depth $\Delta_t^*$, below which these trap-induced losses may be avoided ($\Delta_t < \Delta_t^*$), is given by $\Delta_t^* \approx kT\ln(2qN_{L,A}d/[J_{ph}t_{tr}])$, where $J_{ph}$ is the corresponding photocurrent density. In other words, assuming typical values of $t_{tr} \sim 1$ μs, $N_{L,A} \sim 10^{20}$ cm$^{-3}$, and $d = 100$ nm, to avoid trap-induced losses in organic solar cells the trap depth needs to be smaller than 0.25 eV, while it only needs be smaller than 0.4 eV in indoor cells (operating at 0.3% of 1 sun). This suggests that the associated photocurrent losses in indoor cells, and similar applications, such as photodetectors, operating at low light intensities, may be avoided.

The presence of trap states in these systems is also visible from ultra-sensitive EQE measurements, performed using a recently introduced approach which allows us to probe sub-gap features far below the CT state energy[32]. This is demonstrated for the model system PCDTBT:PC$_{70}$BM in Fig. 3. To confirm that these features are associated with first-order trap-assisted recombination, we intentionally added trace amounts of m-MTDATA to the PCDTBT: PC$_{70}$BM active layer. Here, m-MTDATA will specifically act as a hole trap due to its HOMO energy level of 5.1 eV[33]. As evident from Fig. 3a, b, the degree of trap-assisted recombination and associated QE loss in PCDTBT:PC$_{70}$BM is drastically increased as one would have expected by adding 1% m-MTDATA with respect to PCDTBT weight fraction, corresponding to approximately 0.9% molar ratio. The associated trap depth was estimated to be 0.37 eV, corresponding to a hole trap energy of $E_t \approx 4.9$ eV (assuming HOMO = 5.3 eV for PCDTBT[34]), close to the HOMO level of

Subsequently, and considering the same set of traps, we were able to consistently explain the intensity-dependent features of both EQE and $V_{oc}$. In particular, the presence of (deep) trap states can account for the transitioning ideality factors between 1 and 2 at moderate intensities (cf. Fig. 1c). In fact, these results provide a possible and very plausible explanation for the origin of ideality

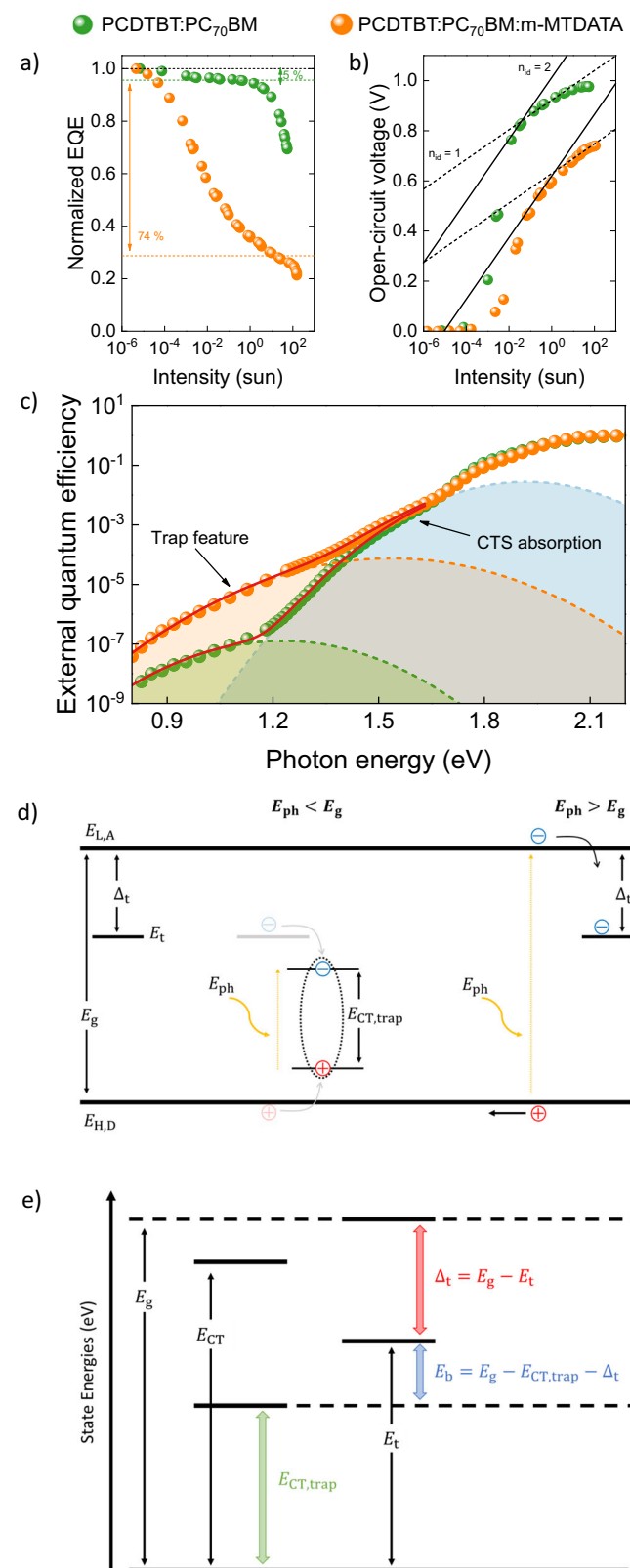

**Fig. 3 Trap states in organic solar cells probed by combined IPC, $V_{oc}$, and ultra-sensitive EQE measurements. a** Normalized light intensity-dependent EQE of a PCDTBT:PC$_{70}$BM solar cell, and compared to a PCDTBT:PC$_{70}$BM:m-MTDATA (1%) device. **b** Open-circuit voltage ($V_{oc}$) of a PCDTBT:PC$_{70}$BM and PCDTBT:PC$_{70}$BM:m-MTDATA solar cells shown as a function of intensity. Dotted and solid lines are guides to the eye corresponding to ideality factors of $n_{id} = 1$ and $n_{id} = 2$ respectively. **c** EQE spectra of PCDTBT:PC$_{70}$BM and PCDTBT:PC$_{70}$BM:m-MTDATA (1%) BHJ solar cells plotted as a function of photon energy. EQEs at photon energies below the gap are fitted with a double Marcus function (solid lines) accounting for both charge transfer state and sub-gap absorption features. **d** Schematic energy level diagram of a donor: acceptor blend with $E_{L,A}$ and $E_{H,D}$ denoting the acceptor LUMO and donor HOMO energy level. The (electron) trap state is located at an energy $E_t$ within the donor: acceptor gap $E_g$. The associated charge generation paths in ultra-sensitive EQE (photoexcitation energy $E_{ph}$ smaller than $E_g$) and IPC (photoexcitation energy $E_{ph}$ larger than $E_g$) measurements are indicated by up and downward arrows. **e** State energies and trap-related parameters in organic solar cells, which are obtained as such from ultra-sensitive EQE ($E_{CT,trap}$) and IPC measurements ($\Delta_t$), are interconnected via the binding energy $E_b$.

The corresponding ultra-sensitive EQE vs. incident light energy is shown in Fig. 3c. Assuming Marcus-type charge-transfer, the low-energy sub-gap feature is expected to correspond to the excitation of a charge-transfer complex between a free and a trapped charge carrier, as recently shown by Zarrabi et al.[20]. The corresponding CT energy of the sub-gap feature in the neat PCDTBT:PC$_{70}$BM is estimated to be $E_{CT,trap} \approx 0.74$ eV (see Supplementary Note 3)[20]. With the trap depth $\Delta_t = 0.48$ eV, and assuming a donor: acceptor effective gap of $E_g = 1.4$ eV for charge-separated states, this corresponds to a binding energy ($E_b = E_g - \Delta_t - E_{CT,trap}$) of 0.18 eV. With the addition of m-MTDATA the sub-gap feature is blue-shifted, consistent with the smaller trap depth ($E_{CT,trap} \sim 0.85$ eV). It is a subtle but important point to note that we used the same thickness for both devices in order to reduce the influence of the interference effects on the sub-gap EQE[35].

These findings suggest that the low-energy sub-gap features observed in the ultra-sensitive EQE measurements are of the same origin as the trap states probed by IPC, with the difference being alternate photoexcitation paths. While in ultra-sensitive EQE measurements the trap states are directly excited (i.e., the photon excitation energy $E_{ph}$ is much lower than $E_g$) forming (coulombically) bound trapped-electron/mobile-hole pairs in the case of electron traps, in IPC measurements the trapping of free, separated, charge carries with excess energy (since $E_{ph} > E_g$) is observed (see Fig. 3d). The two parameters, trap depth $\Delta_t$ (obtained from IPC) and charge-transfer trap energy $E_{CT,trap}$ (obtained from ultra-sensitive EQE measurements), are related via a binding energy $E_b = E_g - \Delta_t - E_{CT,trap}$, which generally leads to a reduction of the trap depth for separated charge carriers (cf. Fig. 3e). We note, however, that the exact value of this binding energy is expected to be strongly influenced by the prevailing energetic disorder[36,37].

From the above analysis, it is unclear whether the observed traps are electron or hole traps. For this purpose, we conducted dark space-charge-limited current (SCLC) measurements on electron-only and hole-only devices (see Supplementary Fig. 5). The SCLC results point towards these traps being predominately acceptor-type electron traps. However, it should be stressed that SCLC is very challenging being highly sensitive to the energetics at the contacts, (unintentional) doping in the bulk, and energetic disorder, complicating the analysis[38–40]. We also note that the traps, and the associated losses, seem to be agnostic to changes in

m-MTDATA. Moreover, the trap density is estimated to increase from $N_t \approx 5 \times 10^{16}$ cm$^{-3}$ in neat PCDTBT:PC$_{70}$BM to $N_t \approx 10^{18}$ cm$^{-3}$ after adding m-MTDATA. Although this is only a rough estimate, the number density of added traps, relative to the total density of available transport sites, is consistent with the added 1% m-MTDATA.

morphology (see Supplementary Note 4). The observed trap depths are, however, consistent with the findings by Zuo et al. In the work by Zuo et al., it was proposed that electron (and hole) traps lying ~0.3–0.4 eV below (or above) the LUMO (HOMO) are induced by dielectric effects from water-filled nanovoids in neat organic semiconductor films[11]. A similar situation is also expected to occur for BHJ structures, pointing towards a universal presence of water-filled nanovoids in BHJs. A detailed morphological structure-property analysis is, however, needed to further confirm this proposition.

## Discussion

In conclusion, we have shown that first-order recombination losses caused by traps are universally present in a large variety of fullerene and non-fullerene acceptor-based organic semiconductor BHJ solar cells. This loss is caused by trap states situated $\sim 0.35 - 0.6$ eV below the transport edges of the acceptor: donor blend having trap densities between $10^{16}$ and $10^{17}$ cm$^{-3}$. The associated trap-assisted recombination not only induces losses in the photocurrent, but also limit the open-circuit voltage giving rise to ideality factors generally between 1 and 2. However, this first-order photocurrent loss is switched off at low intensities (well below 1 sun), suggesting that trap-induced losses may be avoided in related photovoltaic applications operating at lower intensities. Our findings not only shed new light on the nature, dynamics, and role of traps in light-harvesting organic semiconductor devices, but also reveal new insight into the measurement, interpretational complexities, and variability of ideality factors in solar cells. Wide dynamic range, modulated intensity-dependent measurements are a powerful tool in probing the fundamental structure-property relationships in photovoltaic materials.

## Methods

**Device fabrication**. Details to materials and device fabrication are provided in the Supplementary Information.

**Intensity-dependent photocurrent (IPC) and open-circuit voltage ($V_{oc}$)**. IPC and intensity-dependent $V_{oc}$ measurements were performed in air under low frequency (<0.1 Hz) AC modulation using a continuous wave laser with an excitation wavelength of 520 nm to correct for the dark current. A motorized variable two-wheel attenuator (Standa, 10MCWA168-1) with different optical density filters was used to attenuate the incident light irradiance. The external photocurrent and open-circuit voltage of the device under test as well as the photocurrent of the Silicon photodiode (Thorlabs, SM05PD1A) for incident light intensity readings were recorded with Keithley 2450 source-meter-units. The incident light irradiance at the device position was calibrated using a NIST-calibrated Si photodiode power sensor (Thorlabs, S121C).

**External quantum efficiency (EQE)**. EQE measurements were performed in air using a Lambda950 (Perkin Elmer) spectrophotometer as a light source. The output light of the monochromator was chopped at a frequency of 273 Hz (Thorlabs, Thorlabs MC2000B). The photocurrent signal was fed to a current pre-amplifier (Femto, DLPCA-200) before being analyzed with a lock-in amplifier (Stanford Research Systems, SR860). The EQE system was calibrated using a NIST-calibrated silicon (Newport, 818-UV) and germanium (Newport, 818-IR) photodiode sensor. A detailed description of the EQE setup is provided elsewhere[32].

**Current density vs. applied voltage characterization**. The current density vs. applied voltage (JV) characterization was performed in air on encapsulated devices after fabrication. An Oriel LCS-100™ (model 94011A) solar simulator was used in combination with an automated Ossila solar cell I–V test system. The power of the solar simulator was adjusted by using a Newport calibrated reference cell (model 91150 V, series number 2087). An illumination mask with area of 0.0256 cm$^2$ was applied to solar cells with pixel size of 0.04 cm$^2$ for light JV measurements. The area of the mask was determined using an optical microscope. The JV curves were scanned from −1.0 to 1.1 V in the forward direction with 0.02 V step size and scan speed of 0.5 V/s. A Keithley 2400 source-meter-unit was used for measuring dark current density vs. applied voltage [respectively space-charge-limited current (SCLC)] characterizations.

**Resistance-dependent photovoltage (RPV)**. RPV transients were recorded in air with a Rhode & Schwartz Oscilloscope (RTMS3004). Load resistances were varied between 50 W and 1 MW. A Pharos PH1-10 laser with a pulse duration of ~290 fs, excitation wavelength of 532 nm, and repetition rate of 20 Hz was used to generate the charge carriers, while neutral optical density filters were used to attenuate output power to less than 1 nJ/cm$^2$. A detailed description of the RPV setup is provided elsewhere[41].

**Reporting summary**. Further information on experimental design is available in the Nature Research Reporting Summary linked to this paper.

## Data availability

The data that support the findings of this study are available from the corresponding authors upon reasonable request.

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

## Acknowledgements

This work was supported by the Sêr Cymru II Program through the European Regional Development Fund, Welsh European Funding Office, and Swansea University strategic initiative in Sustainable Advanced Materials. A.A. is a Sêr Cymru II Rising Star Fellow and P.M. a Sêr Cymru II National Research Chair. N.Z. and S.Z. are recipients of EPSRC Doctoral Training Program and Swansea University PhD studentships, respectively. This work was also funded by UKRI through the EPSRC Program Grant EP/T028511/1 Application Targeted Integrated Photovoltaics.

## Author contributions

A.A. and P.M. provided the overall leadership of the project. S.Z., O.J.S., and A.A. conceptualized the idea. S.Z. and A.A. designed the experiments. S.Z. measured the light intensity-dependent IPC and $V_{oc}$, J-V, EQE, analyzed the data, and partially fabricated devices. O.J.S. performed the simulations and developed the theory. N.Z. fabricated the devices and partially performed EQE measurements. W.L. fabricated the devices. All authors contributed to the data interpretation and development of the manuscript first drafted by S.Z.

## Competing interests

The authors declare no competing interests.
