## [Peer Review File · Nature Communications]

REVIEWER COMMENTS

Reviewer #1 (Remarks to the Author):

The authors have made an honest effort to address my comments from the previous submission (before transfer). There is now close to complete summary of the model in the SI, but there are still omissions which are crucial to underpinning their conclusions and so I must recommend that this is further amended before I can recommend publication.

At the beginning of the SI section entitled "Drift-diffusion simulations" the five governing equations are presented. This is a 6th order system of equations and so 6 boundary conditions must be supplied in order that a well-posed mathematical problem is being solved. Later in that section 3 boundary conditions are given but a further 3 are still not specified. Of the 3 missing conditions, I think, 2 are conditions on the interfacial recombination rates. The recombination rates are key in determining the shape of the JV curve and so the results of the simulations could not be reproduced, and indeed cannot be meaningfully interpreted, at present. In my opinion, this really needs to be fixed.

Reviewer #2 (Remarks to the Author):

All of my previous comments have been adequately addressed and it is my opinion that the manuscript is well suited to publication in Nature Communications.

Reviewer #3 (Remarks to the Author):

I read both the revised version of the manuscript and the rebuttal letter. In doing so, I noticed that the changes to the manuscript are minimal, in stark contrast with the large number of concerns raised by all reviewers, some of them major, and that seriously question the validity of the conclusions. The authors have indeed answered each question in the response letter, but they mostly attempted to justify their initial work and have not incorporated the concerns into the revisions made to the manuscript. My most serious concerns related to this revised manuscript are as follows:

1. the novelty of the results (not clear because of prior work, as already mentioned);
2. the technical approach on which the conclusions are based: with such a large number of variables, any experimental results can be modelled to support practically any theory;
3. The work was carried on devices that are significantly inferior in performance with state of the art OPV. The processes occurring in these unoptimized devices are most likely not relevant to the operation of the high performance OPV and hence, the impact of the work is not clear.

Overall, the work is not of sufficient quality, novelty and impact to grant publication in Nat Commun.

Response Letter

We would like to thank the reviewers for careful reading of our work as well as providing helpful comments. In the following we have listed a detailed point-by-point response to all the Reviewers' comments. Our response is in **Blue** and corresponding changes in the revised manuscript have been indicated by **Red**.

Reviewer #1 (Remarks to the Author):

The authors have made an honest effort to address my comments from the previous submission (before transfer). There is now close to complete summary of the model in the SI, but there are still omissions which are crucial to underpinning their conclusions and so I must recommend that this is further amended before I can recommend publication.

At the beginning of the SI section entitled "Drift-diffusion simulations" the five governing equations are presented. This is a 6th order system of equations and so 6 boundary conditions must be supplied in order that a well-posed mathematical problem is being solved. Later in that section 3 boundary conditions are given but a further 3 are still not specified. Of the 3 missing conditions, I think, 2 are conditions on the interfacial recombination rates. The recombination rates are key in determining the shape of the JV curve and so the results of the simulations could not be reproduced, and indeed cannot be meaningfully interpreted, at present. In my opinion, this really needs to be fixed.

Response: We thank the reviewer for their comments. We have now revised the SI section accordingly. All 6 boundary conditions are now included in the SI.

Furthermore, to better account for the prevailing recombination rates (in the bulk and at the electrodes) we have now also included realistic generation profiles into the simulations, noting that the profiles under intensity-dependent measurements (520 nm laser) and 1 sun conditions are different. The generation profiles are now calculated based on an optical transfer-matrix model, using the measured complex refractive indices as input. In addition, we have revisited the IPC measurement set-up to minimize measurement-related errors. Subsequently, we repeated all measurements on fresh solar cells. As a result of these changes, the simulations are now able to simultaneously reproduce the experimental intensity-dependent EQE and the open-circuit voltage, as well as the measured JV curves. We feel this to be a major improvement.

Reviewer #2 (Remarks to the Author):

All of my previous comments have been adequately addressed and it is my opinion that the manuscript is well suited to publication in Nature Communications.

Response: We thank the reviewer for their positive comments.

Reviewer #3 (Remarks to the Author):

I read both the revised version of the manuscript and the rebuttal letter. In doing so, I noticed that the changes to the manuscript are minimal, in stark contrast with the large number of concerns raised by

all reviewers, some of them major, and that seriously question the validity of the conclusions. The authors have indeed answered each question in the response letter, but they mostly attempted to justify their initial work and have not incorporated the concerns into the revisions made to the manuscript.

Response: We thank the reviewer for their insightful comments. To address the reviewer's comments and concerns, we have thoroughly revisited all experiments and simulations and updated the manuscript and Supplementary Information accordingly. The detailed responses to the comments are given below.

My most serious concerns related to this revised manuscript are as follows:

1. the novelty of the results (not clear because of prior work, as already mentioned);

Response: In the revised manuscript, we have now made changes to better reflect the novelty points. In particular, we have added a section about the requirements for trap depth to avoid trap-induced losses depending on the operating light intensity. State-of-the-art systems are included in the revised version. Further, and as mentioned in our previous reply, the relevance and presence of first-order trap-assisted recombination in OPVs is still debated. Below, we have listed reasons for why we think our work is novel and of general interest to the community:

-Previously, trap-assisted recombination has been mainly inferred from measurements of the ideality factor or transient techniques, but as we point out in the Introduction this approach is unreliable due to limitations inherent to the experimental open circuit conditions such as shunt resistance. Traps, in turn, have mainly been detected via space-charge-limited current measurements, however, these are very challenging in practice and not directly applicable to operational devices. In this work, we present a new method which enables us to estimate both the trap depth and the first-order recombination in organic solar cells – this has not been possible in operational devices before. Our technique is based on a feature that is unique to SRH recombination, namely the two-step EQE, which allows us to unambiguously detect its presence.

-Furthermore, with this technique it is possible to differentiate between first-order trap-assisted recombination in the bulk and other first-order losses at short-circuit. This has not been possible before.

-Subsequently, we were able to unambiguously detect the universal presence of trap-assisted recombination in organic solar cells, resulting in actual measurable photocurrent loss at short-circuit (this is opposed to previous works which have focused on open-circuit conditions where actual losses are difficult to estimate).

-We find that the trap depth of these traps is universally around 0.4-0.6 eV below the gap in all organic solar cells tested, independent of whether these are fullerene or non-fullerene-based blends. Previously, little has been known about the trap depth in organic solar cells and the debate has mainly centred around whether the traps are mid-gap states or tail states directly below the band edge. In our work, we show, for the first time, that the relevant traps in OPVs are neither but rather composed of “semi-deep” states that are too deep to be tail states yet too shallow to be mid-gap states.

- We also observe that the trap-assisted recombination in the bulk can be deactivated at lower intensities (manifest by the two-step EQE). This is direct consequence of the “semi-deep” traps and, to our knowledge, has not been observed before in organic solar cells. This has direct implications for other photovoltaic applications, such as indoor solar harvesting and photodetectors operating under

low illumination conditions. In fact, given that indoor cells operate at intensities 0.1-1% of 1 sun, our results suggest that these losses may be avoided in such applications.

-Additionally, the insight that the onset light intensity, at which the trap-assisted recombination is activated, is directly dependent on the trap depth can be used as a “design rule” for how to avoid trap-induced losses in photovoltaic applications operating at different intensities. For example, a quick back-of-the-envelope calculation suggests that, to avoid trap-induced losses, the trap depth needs to be smaller than 0.25 eV in organic solar cells at 1 sun, while it only needs to be smaller than 0.4 eV for indoor cells.

-Finally, we also provide some insight into the relation between trapped charge carriers and the associated CT complex between trapped and free charge carriers, which has not been discussed before in the context of traps in organic solar cells.

This said, we appreciate the comment and have made appropriate and substantial changes to reflect the novelty as stated above.

2. the technical approach on which the conclusions are based: with such a large number of variables, any experimental results can be modelled to support practically any theory;

Response: We would like to note that if the theory does not contain the critical key features of the experimental data, it cannot possibly model the observed features (such as two-step EQE) in the first place, no matter the number of free parameters. Furthermore, we would like to clarify on the point that a relatively large number of parameters only enters when directly reproducing the experimental data with numerical simulations; however, the only purpose of these simulations was to confirm the experimental observation and not to fit them.

To address the Reviewer’s concern, in the revised manuscript, we have carefully repeated the experiments and revisited the numerical simulations to minimize the number of free variables (see below). As a result, the simulations are now able to reproduce not only the experimental intensity-dependence of EQE and Voc, but also the experimental JV curves. However, we would like to reiterate the fact that the conclusions of this paper are NOT based on fitting experimental data to a model with a large number of variables.

In fact, our results can be summarized as follows: i) we detect the presence of a first-order recombination loss activated at moderate intensities (but still < 1 sun) under short-circuit conditions; ii) based on this experimental observation we suggest that this first-order loss is induced by trap-assisted recombination; iii) from the onset of this first-order loss, we estimate the trap depth; iv) we use drift-diffusion simulations to check the consistency of our conclusions; and finally v) we conduct other independent experiments that further support our conclusions. By further examining these points (i-v), we can deduce the following:

-The first result, (i), that an additional first-order recombination is activated at moderate intensities, and is completely agnostic to any specific theory or model. An EQE loss that is independent of intensity is, by definition, a first-order photocurrent loss with respect to intensity. The fact that the experimental alpha is very close to unity supports this observation.

-Next, in (ii), we find that trap-assisted recombination, based on the well-established (and very general) SRH theory, can explain the two-step behaviour of the intensity-dependent EQE behaviour. In fact, the two-step behaviour is a key feature of SRH kinetics itself, independent of parameters or

assumed transport model. Here, it is very important to stress that the SRH theory is not just one theory among many candidates but, it is the only viable recombination mechanism or theory that can consistently explain the (experimentally observed) deactivation of the additional first-order recombination loss at low intensities. A two-step EQE behaviour cannot be reproduced with a drift-diffusion model that lacks SRH recombination (the “no traps” case), regardless of the number of free parameters. The presence of SRH is further supported by the experimentally observed ideality factors of two of the Voc, but also by sensitive EQE measurements.

-Then, in (iii), based on the SRH theory, we find a simple analytical model to determine the trap depth from the point-of-transition between the two EQE steps. However, this trap depth determination only requires a few parameters, all of which are experimentally estimated. Importantly, the trap depth depends only weakly (logarithmically) on these parameters; hence, an order of magnitude uncertainty within these parameters are still within the indicated error bars for the estimated trap depth. Furthermore, as demonstrated in Figure S8, an excellent agreement between the simple analytical model (used for trap depth determination) and the full numerical drift-diffusion simulations is obtained, even when varying several different parameters over several orders of magnitude. This means that the simple analytical model remains valid and independent of all other parameters in the numerical drift-diffusion simulations; hence, the trap depth determination does not rely on any free parameters.

-It is only during (iv), when a numerical drift-diffusion model is used to reproduce the experimental data, that a relatively large number of parameters is required. We stress that the drift-diffusion simulations are only used to support the data and prove the consistency – our conclusions do not rely on these. However, to reduce the number of variables, we have now also accounted for the generation profile in the active layer (which depends on whether 1 sun conditions or the IPC measured at 520 nm are considered). To this end, an optical transfer-matrix model which uses experimentally determined complex refractive indices as input is utilized. Furthermore, we have carefully revisited the measurements on fresh devices, taking precautions to minimize effects of degradation and measurement-related errors. A testament of these improvements is a consistent series resistance (being equal for all devices). Subsequently, the drift-diffusion model is now also able to reproduce both the intensity-dependent EQE and Voc measurements as well as the experimental JV curves. It is important to highlight that the most critical parameter determining the position of the two-step is the trap depth, while the rest of the trap parameters will not affect this position. Although the trap density and capture coefficients are still correlated, their product (determining the SRH lifetime) remains unchanged, and so does the degree of SRH recombination. Furthermore, most of the parameters dominate in different intensity regimes. For example, the series resistance and bimolecular recombination rate constant only define the highest intensity regime (where the EQE drops quickly), having no influence on the lower intensity regimes. Conversely, the shunt resistance only determines the shunt-dominated Voc regime at low intensities. Hence, even though the number of parameters seem large, the number of free parameters that operates simultaneously within the same intensity regime is small.

-Finally (v), it should be noted that we also provide several other experimental findings supporting our conclusions, independent of model or simulations.

3. The work was carried on devices that are significantly inferior in performance with state of the art OPV. The processes occurring in these unoptimized devices are most likely not relevant to the operation of the high performance OPV and hence, the impact of the work is not clear.

Response: This is indeed true, and we very much appreciate the comment. In the revised manuscript, we have now included state-of-the-art systems, most notably PM6:Y6 and the even higher-performing PM6:BTP-eC9 systems. These two non-fullerene systems both exhibit PCEs >15 %. Furthermore, we have thoroughly and carefully repeated all the measurements on fresh solar cells and taken precautions to avoid degradation. Based on these measurements, first-order trap-assisted quantum efficiency losses are found to be between 3 % (PTB7-Th:ITIC, PM6:Y6 and PM6:BTP-eC9) and 10 % (PBDB-T:IT-4F). Importantly, our conclusions remain unchanged and agree with our previous set of devices. These new results unambiguously show the relevance of our work for OPVs in general, and NFA-based, state-of-the-art organic solar cells.

Changes made to the MS:

All previous experimental data and corresponding simulations have been replaced with new ones conducted on state-of-the-art OPV cells, constituting of both fullerene and non-fullerene acceptor-based cells. 14 new OPVs have been fabricated, measured, and analysed. The device fabrication section, Fig. S1-6,8 and 9, Supplementary Note 4 as well as Table T1 and T2 in the Supporting Information have been revised accordingly. Fig. 2 and 3, names of materials, numbers of obtained trap depths, densities and QE losses in the main text have also been revised accordingly.

Furthermore, we have amended the sentences between line 19 and 24 in the abstract to read as follows:

“The trap-assisted recombination is found to be induced by states lying 0.35-0.6 eV below the transport edge, **acting as deep trap states at light intensities equivalent to 1 sun**. Apart from limiting the photocurrent, we show that the associated trap-assisted recombination *via* these **comparatively deep traps** is also responsible for ideality factors between 1 and 2, shedding further light on another open and important question as the fundamental working principles of organic solar cells. **Our results also provide insights for avoiding trap-induced losses in related indoor photovoltaic and photodetector applications.**”

We have added a sentence in line 60 in the main text to read as follows:

“While the debate has been heavily centred around whether these traps are mid-gap states or shallow tail states below the transport level, surprisingly little is known about their energetics.”

We have added a sentence in line 74 in the main text to read as follows:

“The two-step behaviour is a result of trap-induced first-order recombination in the bulk (which is absent at low intensities) being switched on at moderate intensities due to trap filling.”

We have amended the sentence in line 77 in the main text to read as follows:

“This effect allows for the first-order trap-assisted recombination loss **under 1 sun intensity** to be quantified.”

We have added a sentence in line 83 in the main text to read as follows:

“Finally, our findings are also relevant for indoor solar harvesting and photodetector applications of organic BHJs operating at low intensities.”

We have amended the sentence in line 100 in the main text to read as follows:

“Under conditions when $E_{F,n} \ll E_t$ (low I_L , see inset in **Fig. 1b**), most of the traps are unoccupied (**shallow trap mode**), and the trap-assisted recombination is negligible (compared to charge extraction).”

We have amended the sentence in line 102 in the main text to read as follows:

“In contrast, when $E_{F,n} \gg E_t$ (high I_L , see inset in **Fig. 1b**), a significant fraction of traps will be occupied by electrons (**deep trap mode**), and first-order trap-assisted recombination is switched on, resulting in the second EQE plateau.”

We have added a sentence in line 109 in the main text to read as follows:

“Hence, the trap depth critically defines the onset intensity for first-order SRH recombination in the bulk – below this onset, the traps are acting as shallow traps and the trap-induced first-order recombination losses in the bulk is small.”

Furthermore, we have amended the sentence in line 136 in the main text to read as follows:

“Fig. 2a and b show the light-intensity-dependent normalized EQE and V_{oc} , respectively, of three different organic solar cells: **PCDTBT:PC₇₀BM**, **PTB7-Th:PC₇₀BM** and **PM6:BTP-eC9**.”

We have amended the sentence in line 140 in the main text to read as follows:

“Corresponding first-order, trap-induced, relative QE losses of **5 %**, **4 %** and **3 %** were obtained from the IPC measurements for the three systems, respectively.”

We have added the sentence in line 141 in the main text to read as follows:

“However, this loss channel is deactivated below intensities around 10^{-3} - 10^{-4} suns for PCDTBT:PC₇₀BM and PTB7-Th:PC₇₀BM, and 10^{-2} suns for PM6:BTP-eC9.”

We have amended the sentence in line 148 in the main text to read as follows:

“This *two-step* EQE behaviour was detected for a large variety of **fullerene and non-fullerene acceptor based organic solar cells (see Fig. S4)**.”

We have amended the sentence in line 172 in the main text to read as follows:

“The experimental EQE and V_{oc} (**and corresponding J-V curves under 1 sun**) can be reproduced qualitatively by drift-diffusion simulations, assuming a device with trap states lying 0.4-0.5 eV below the transport levels and a finite shunt resistance, as indicated by the solid lines in Fig 2a and b (**and Fig. S9**).”

We have added a paragraph in line 184 in the main text to read as follows:

“Our results are consistent with the presence of trap-assisted recombination that is activated at moderate intensities by trap filling of states, lying 0.35-0.6 eV below the transport levels, acting as deep trap states under 1 sun illumination. The onset intensity, at which the first-order trap-assisted recombination in the bulk is activated, is determined by the trap depth. Concomitantly, the critical trap depth Δ_t^* , below which these trap-induced losses may be avoided ($\Delta_t < \Delta_t^*$), is given by $\Delta_t^* \approx kT \ln(2qN_{L,A}d/[J_{ph}t_{tr}])$, where J_{ph} is the corresponding photocurrent density. In other words, assuming typical values of $t_{tr} \sim 1 \mu s$, $N_{L,A} \sim 10^{20} \text{ cm}^{-3}$, and $d = 100 \text{ nm}$, to avoid trap-induced losses in organic solar cells the trap depth needs to be smaller than 0.25 eV, while it only needs be smaller than 0.4 eV in indoor cells (operating at 0.3% of 1 sun). This suggests that the associated photocurrent

losses in indoor cells, and similar applications such as photodetectors, operating at low light intensities, may be avoided.”

We have amended the sentences in line 203-207 in the main text to read as follows:

“The associated trap depth was estimated to be **0.48 eV**, corresponding to a hole trap energy of $E_t \approx 4.8 \text{ eV}$ (assuming HOMO = 5.3 eV for PCDTBT³²), close to the HOMO level of m-MTDATA. Moreover, the trap density is estimated to increase by a factor of **10** from $N_{t,\text{neat}} \approx 5 \times 10^{16} \text{ cm}^{-3}$ in neat PCDTBT:PC₇₀BM to $N_{t,\text{added}} \approx 10^{18} \text{ cm}^{-3}$ after adding m-MTDATA.”

We have amended the sentence in line 215 in the main text to read as follows:

“With the trap depth $\Delta_t = 0.48 \text{ eV}$, and assuming a donor: acceptor effective gap of $E_g = 1.4 \text{ eV}$ for charge-separated states, this corresponds to a binding energy ($E_b = E_g - \Delta_t - E_{CT,\text{trap}}$) of **0.18 eV**.”

We have added the sentence in line 255 in the conclusion to read as follows:

“However, this first-order photocurrent loss is switched off at low intensities (well below 1 sun), suggesting that trap-induced losses may be avoided in related photovoltaic applications operating at lower intensities.”

We have amended the sentence in line 362 in the caption of Figure 1 to read as follows:

“**a**) Experimental (blue symbols) and simulated (red lines) normalized external quantum efficiency (EQE) plotted as a function of intensity for three different organic solar cells together with the estimated relative QE losses induced by first-order, trap-assisted recombination: **PCDTBT:PC₇₀BM (5 %), PTB7-Th:PC₇₀BM (4 %) and PM6:BTP-eC9 (3 %).**”

REVIEWERS' COMMENTS

Reviewer #1 (Remarks to the Author):

The authors have addressed my comments. I am now satisfied with the work.

Reviewer #3 (Remarks to the Author):

The manuscript has been improved significantly and my main concerns have been addressed adequately. I recommend publication, but I advise the authors to revise Figure 2 because it is almost impossible to read the text in it without significant zoom in.